# Evaluation of Dry Ice for Short-Term Storage and Transportation of Frozen Boar Semen

**DOI:** 10.3390/ani14101422

**Published:** 2024-05-10

**Authors:** Mengqian He, Lingwei Sun, Jiehuan Xu, Caifeng Wu, Shushan Zhang, Jun Gao, Defu Zhang, Yeqing Gan, Yi Bian, Jinliang Wei, Weijian Zhang, Wengang Zhang, Xuejun Han, Jianjun Dai

**Affiliations:** 1Shanghai Municipal Key Laboratory of Agri-Genetics and Breeding, Institute of Animal Husbandry and Veterinary Science, Shanghai Academy of Agricultural Sciences, Shanghai 201106, China; sunlingwei@126.com (L.S.); jiehuanxu810@163.com (J.X.); wucaifengwcf@163.com (C.W.); zhangshushan@saas.sh.cn (S.Z.); gjsaas@gmail.com (J.G.); zhangdefuzdf@163.com (D.Z.); 2Key Laboratory of Livestock and Poultry Resources (Pig) Evaluation and Utilization, Ministry of Agriculture and Rural Affairs, Shanghai 201106, China; 3Shanghai Engineering Research Center of Breeding Pig, Shanghai 201106, China; 13887223118@163.com; 4Shanghai Jiading Municipal Centre for Disease Control and Prevention, Shanghai 201899, China; 18930381338@163.com (Y.G.); kytdyx2022@163.com (Y.B.); 13671798163@163.com (J.W.); 5Shanghai Municipal Centre for Disease Control and Prevention, Shanghai 200051, China; xumuke021@163.com (W.Z.); eligaori@163.com (W.Z.)

**Keywords:** semen cryopreservation, dry ice, semen transportation, boar

## Abstract

**Simple Summary:**

Liquid nitrogen (LN) is usually used for semen cryopreservation. The most significant point is that it has no time or space restrictions, maximizing the use of high-quality males for breeding. Dry ice is also one of the earliest media used in animal semen cryopreservation. The efficiency of dry ice cryopreservation is lower than that of LN. However, compared with LN, dry ice has an advantage in transportation, as it permits express delivery and can be carried on trains and high-speed railways. Therefore, using dry ice as a cold source for preserving and transporting ultralow-temperature-preserved semen liquid has received extensive attention from scholars. It has been demonstrated in frozen semen from mice and cattle that dry ice can be used as a reliable cold source for transportation. For this reason, we simulated a dry ice transportation environment by placing ultralow-temperature-frozen semen in a foam box with sufficient dry ice to maintain it for one week. We found that one week of storage in dry ice did not cause significant damage to porcine spermatozoa. The effects on the semen were essentially the same as those of LN storage when the dry-ice-preserved spermatozoa were transferred to LN again after one week.

**Abstract:**

To address the safety problems posed by the transportation of boar semen using LN, this study was conducted on the short-term storage of frozen boar semen in dry ice (−79 °C). Boar semen frozen in LN was transferred to dry ice, kept for 1 day, 3 days, 5 days, 7 days, or 8 days, and then moved back to LN. The quality of frozen semen stored in LN or dry ice was determined to evaluate the feasibility of short-distance transportation with dry ice. The results showed that 60 °C for 8 s was the best condition for thawing frozen semen stored in dry ice. No significant differences in spermatozoa motility, plasma membrane integrity, or acrosome integrity were observed in semen after short-term storage in dry ice compared to LN (*p* > 0.05). There were no significant changes in antioxidant properties between storage groups either (*p* > 0.05). In conclusion, dry ice could be used as a cold source for the short-term transportation of frozen boar semen for at least 7 days, without affecting sperm motility, morphological integrity, or antioxidant indices.

## 1. Introduction

Livestock germplasm resources are strategic resources for a country and form the basis of future breeding efforts. With the implementation of “seed industry vitalization” in China, collecting and protecting local breed characteristics and the gametes of endangered species has become a priority at this stage and will continue to be a priority in the long term. Methods for protecting the genetic resources of livestock and poultry include living and static preservation. Static preservation includes the cryopreservation of semen, embryos, oocytes, DNA, and somatic cells. Ultralow-temperature preservation keeps spermatozoa in a relatively quiescent state to achieve a more efficient protection of germplasm resources [1]. Frozen semen has been widely used in assisted reproduction, which has allowed the widespread use and preservation of genetically superior semen [2]. Due to the development of semen banking, which began in 2002, frozen semen was proven to be an effective and safe assisted reproduction technology, and it has a similar fertilization rate to fresh semen [3]. Thus, frozen semen is an effective alternative to live animals. The use of semen-freezing technology means that semen can be treated using a particular method and preserved in a dormant state in an ultralow-temperature environment.

Moreover, after thawing, the structure of spermatozoa is still intact and can be used for fertilization [4]. Recent research on boar semen cryopreservation technology has paid increasing attention to livestock production [5]. Frozen boar semen preservation technology is of great significance to the development of the boar industry. Frozen semen can cross the limitations of time and space and significantly improve the utilization rate of high-quality male animals. Meanwhile, it can achieve the long-term preservation of excellent germplasm resources, which is significant in protecting local genetic resources and collecting genetic data.

Frozen semen is generally preserved using LN as the cold source. At −196 °C, the biochemical and photodynamic reactions of semen are effectively inhibited, which is optimal for long-term storage [6]. LN belongs to the second category of compressed and liquefied gases, which are prone to leakage during transportation if not correctly sealed, adversely affecting the samples. It is suitable for preserving frozen semen, but not for long-distance transportation. Dry ice (−79 °C) is also a common cold source and one of the earliest media applied to the cryopreservation of animal semen. There were many concerns that the efficiency of cryopreservation with dry ice might not be as good as that with LN. Dry ice is solid and can be transported in foam boxes, and is an excellent cold source for frozen goods. Although the refrigeration effect of dry ice is not as perfect as LN, it can still be achieved using short-distance vehicles. Thus, in many cases, it is more challenging to transport semen using dry ice.

In addition to changes in temperature during freezing, boar spermatozoa are also very sensitive to low temperatures. Cryopreservation exacerbates spermatozoon cell stress, damaging cellular structure and function [7], which worsens sperm motility performance. The effects on semen of short-term storage using dry ice vary between species. It was shown that frozen cow semen stored in dry ice for 6 months did not differ significantly in viability from that stored in LN after thawing [8]. Frozen semen from mice could also be kept in dry ice for 7 days, and no significant damage was found when transferred back into LN [9]. However, frozen human spermatozoa stored in dry ice for 48 h showed a significant decrease in viability and mitochondrial activity [10]. So far, a reliable basis for short-term preservation and transportation in dry ice has not been well explored in boar semen. This study therefore investigated the feasibility of dry ice for the short-term storage and transportation of frozen boar semen.

## 2. Materials

### 2.1. Semen Samples

In all experiments, the animal procedures were approved by the Ethics Committee on the Use of Animals of Shanghai Academy of Agricultural Sciences, China (approval ID: SAASPZ0522057). All materials were acquired from Sigma Chemical Co. (St. Louis, MO, USA) unless mentioned otherwise.

For this study, ejaculates were collected using the gloved-hand method from 6 healthy and fertile Shanghai white boars. The samples were then frozen. All of the animals used for this study were in the age range of 30~36 months and were housed at the Shanghai Academy of Agricultural Sciences facilities under a controlled environment. The animals were fed isoprotein and isoenergetic diets according to their breed. All ejaculates underwent the same processing before freezing in terms of collection and transportation. Briefly, after collection, the ejaculates were extended and cooled to 17 °C in BTS (1:1, *v*/*v*). Once this temperature was reached, the semen was transported in a holding tank (17 °C) to the laboratory for further treatment. Only ejaculates with a sperm count ≥200 × 10^6^ sperm/mL, ≥85% of sperm with normal morphology, and ≥75% and ≥80% of motile and viable sperm, respectively, were selected for cryopreservation. Only one sperm sample per boar was cryopreserved using the straw-freezing procedure initially described by [11] and later modified by [12,13]. Briefly, ejaculate was obtained using the gloved method and diluted with BTS (*v*/*v*). The samples were centrifuged at 15 °C for 3 min at 2400× *g*. The residue was then placed in LEY to achieve a cell concentration of 1.5 × 10^9^ cells/mL. After further cooling to 5 °C over 120 min, the diluted spermatozoa were resuspended in LEY-glycerol-Orvus ES Paste (LEYGO) extender (92.5% LEY, 1.5% Equex STM (Nova Chemical Sales Inc., Scituate, MA, USA) and 6% glycerol (*v*/*v*) (pH 6.2 and 1650 ± 15 mOsm/kg) was added to the spermatozoa to obtain a final concentration of 1 × 10^9^ cells/mL. The resuspended and cooled spermatozoa were loaded into 0.5 mL plastic straws. The temperature was adjusted from 4 °C to 1 °C at 1.5 °C for 2 min, from 1 °C to −140 °C at 30 °C for 4.8 min, and then held at −140 °C for 10 min before being transferred to LN for storage.

### 2.2. Chemicals

All materials were acquired from Sigma Chemical Co. (St. Louis, MO, USA) unless mentioned otherwise. The FITC-PNA kit and the rhodamine kit were purchased from Shanghai Source Leaf Biologicals (Shanghai, China), and the acridine orange (AO) kit was purchased from Beijing Solarbio (Beijing, China).

### 2.3. Experimental Design

#### 2.3.1. Semen Freezing

In this experiment, ejaculate was collected from six boars (all the boars were 30~36 months old) and 23 frozen semen tubes were collected from each individual (Figure 1). Three fine tubes of semen were left in LN (group d0), and their quality was examined. Then, the remaining 120 semen tubes were transferred to plastic foam boxes filled with dry ice and tested for sperm viability on days 0, 1, 3, 5, and 7. At least 3 technical replicates were performed for each program. After day 7, the remaining frozen semen tubes were transferred to LN to test sperm survival on day 8 (D8).

The sperm was thawed at 60 °C for 12 s; however, we also took into account the temperature difference between sperm in dry ice and LN. The experiment first explored the thawing time of semen stored in dry ice.

Thawing in dry ice: Remove the semen tubes from dry ice, quickly place them into a 60 °C water bath, and thaw for 6 s, 7 s, 8 s, 9 s, 10 s, and 12 s, respectively. Then, place them into a 37 °C water bath for 10 min after thawing and diluting them 1:9. Following this, the tests of relevant indices can be carried out.

#### 2.3.2. Frozen Semen Motility Test

Semen was evaluated using computer-assisted semen analysis (CASA, IVOS Sperm Analyzer, Hamilton Thorn, Beverly, MA, USA). An amount of 10 µL of semen sample was positioned on a warm (37 °C) glass slide for evaluation. Three fields with at least 200 sperm were randomly selected for observation using a phase-contrast microscope at 400× magnification. Software settings were adjusted for boar semen and the parameters were evaluated as described by Olivares et al. [14].

#### 2.3.3. Plasma Membrane Integrity

Spermatozoa plasma membrane integrity was determined using the hypotonic swelling test (HOST). The tails of normal spermatozoa swelled and curved when exposed to HOS solution, while low-viability or dead spermatozoa did not undergo any significant swelling [15]. In brief, 10 µL semen samples was added to 100 µL of HOST solution (100 mOsm/L, 57.6 mM fructose, and 19.2 mM sodium citrate) at 37 °C for 30 min. Then, 5 µL mixed sample was placed on a preheated slide and covered with a cover slide. The samples were observed in three randomly selected visual fields under an optical microscope. Spermatozoa with coiled or swollen tails were considered HOS+. Semen plasma membrane intactness = number of spermatozoa with curved tails/total number of spermatozoa counted × 100%.

#### 2.3.4. Acrosome Integrity

The evaluation of acrosome integrity was conducted through staining with fluorescein isothiocyanate-labeled peanut agglutinin (FITC-PNA) [16]. The slides were evenly coated with a solution containing FITC-PNA (100 μg/mL in PBS). The slides were incubated in a wet box and incubated for 30 min in the dark. Afterwards, they were washed three times with PBS. At least 200 spermatozoa were observed under a fluorescence microscope. Spermatozoa acrosome integrity rate = spermatozoa with intact acrosomes/total spermatozoa counted × 100%.

#### 2.3.5. DNA Integrity

Acridine orange (AO) fluorescent staining was used to detect semen DNA integrity [17]. After thawing, semen density was adjusted to a final concentration of 5 × 10^6^ mL^−1^. An amount of 10 μL of semen was taken for coating, and then the slides were air-dried and fixed with anhydrous ethanol/glacial acetic acid (V:V = 3:1) for 3 h. The slides were washed thrice with PBS, and then the AO staining solution was added dropwise before incubating the slides in a dark environment for 10 min. The slides were removed and rinsed with PBS to remove excess staining solution and dried naturally in a dark environment. At least 200 spermatozoa were observed under a fluorescence microscope with excitation and emission wavelengths of 480 nm and 530 nm, respectively. Spermatozoa with intact DNA will emit green fluorescence, and those with damaged DNA will emit yellow or red fluorescence. Spermatozoa DNA integrity = DNA intact- spermatozoa/total spermatozoa counted × 100%.

#### 2.3.6. Mitochondrial Activity

Spermatozoa mitochondrial integrity was assessed using a combination of rhodamine 123 (R 123) and propidium iodide (PI) fluorescent staining [18]. After thawing, semen density was adjusted to a final concentration of 5 × 10^6^ mL^−1^. An amount of 100 µL diluted semen was mixed with 1 µL (0.5 mM) rhodamine solution and 1 µL PI (3 mM) and incubated for 30 min in the dark. The samples were washed 3 times with PBS, and 10 µL smears of the semen sample were prepared. No fewer than 200 spermatozoa were observed under a fluorescence microscope with excitation and emission wavelengths of 480 nm and 513 nm, respectively. Red fluorescence in the nuclear region of the head and no green fluorescence in the tail indicate that the spermatozoon is dead and that there is no mitochondrial activity. Red fluorescence or no fluorescence in the nuclear region of the head and green fluorescence in the tail indicate that the spermatozoon is dead or alive, but there is mitochondrial activity. Spermatozoa mitochondrial activity = fluorescent tail/total spermatozoa counted × 100%.

#### 2.3.7. Antioxidant Properties

A total antioxidant capacity (T-AOC) kit was purchased from Nanjing Jiancheng Bioengineering Research Institute (Nanjing, China). Malondialdehyde (MDA), superoxide dismutase (SOD), and reactive oxygen species (ROS) kits were purchased from Biyuntian (Shanghai, China).

The thiobarbituric acid-reactive substance (TBARS) assay was used to measure MDA levels. After thawing, semen density was adjusted to a final concentration of 5 × 10^6^ mL^−1^. Briefly, 1 mL of 15% (*w*/*v*) trichloroacetic acid was added to each tube, followed by 1 mL of 0.375% (*w*/*v*) thiobarbituric acid. The sample tubes were vortexed and then boiled in a water bath for 10 min. Next, the samples were removed from the wastewater and placed in an icebox to stop the reaction. Finally, the supernatant was collected and the absorbance was measured at 532 nm using a Shimadzu UV 2100 spectrophotometer (Shimadzu, Kyoto, Japan). The concentration of MDA was determined using a standard curve containing a known concentration (0.5–32 µM) of MDA [19].

The ROS levels in sperm were determined using 2′,7′-dichlorodihydrofluorescein diacetate (DCFH-DA; Beyotime, Shanghai, China). First, each semen sample was centrifuged at 800 r/min for 5 min. Then, the semen was resuspended with 10 mM DCFH-DA, diluted to a concentration of 5 × 10^6^ spermatozoa/mL, and incubated at 37 °C in the dark. The fluorescence intensity in each sample was assessed every minute for 80 min using a luminescence instrument (BioTek, Winooski, VT, USA). The results are shown as relative fluorescence units (RFUs).

The total antioxidant capacity (TAC) was measured using a colorimetric total antioxidant capacity assay kit (Thermo Fisher Scientific, Waltham, MA, USA) following the manufacturer’s instructions. TAC was measured by monitoring hydrogen peroxide decomposition. First, each semen sample was centrifuged at 20,000× *g* for 10 min. Then, the supernatant was collected and diluted in double-distilled water (1:100). An amount of 100 µL of diluted samples and standards was added to a 96-well plate. All standards and samples were mixed with 100 µL of Cu^®2+^ solution and incubated at room temperature for 90 min. After incubation, the samples were measured immediately in a microplate reader (SPECTROstar Nano, BMG LABTECH, Ortenberg, Germany) at 570 nm. For the colorimetric assay, a 1 mM Trolox standard was prepared and serially diluted for the standard curve, and the concentration was calculated using the calibration curves.

A SOD assay kit (Nasdox™—Superoxide Dismutase Assay Kit, Navand Salamat Company, Urmia, Iran) and the nitro blue tetrazolium reduction method were used to evaluate SOD activity, expressed as U/mL. Sample concentrations (U/mg) were determined by measuring the mean absorbance values with a spectrophotometer at 630 nm and converting the SOD activity using a standard calibration curve.

### 2.4. Statistical Analyses

The data were analyzed using IBM SPSS Statistics 20.0 (SPSS Inc., Chicago, IL, USA). All data were checked using the Shapiro–Wilk test and were found to fit the normal distribution. The groups were compared using one-way ANOVA, followed by post hoc analysis and the least significant difference test. Differences between two groups were analyzed using Student’s *t*-test. The significance level was set to 0.05. Data are presented as mean value ± standard error of the mean (SEM).

## 3. Results

### 3.1. Effect of Different Thawing Times on Semen Quality When Stored on Dry Ice

When semen is stored in LN, it is usually thawed at 60 °C for 12 s. When the semen was transferred into dry ice, it was found that after thawing under these conditions, the spermatozoa survival rate was significantly lower than that of the other groups (*p* < 0.05). This can be seen in Table 1. A comparison of different times at 60 °C showed that thawing times of 6 s, 7 s, and 10 s resulted in significantly (*p* < 0.05) lower spermatozoa motility relative to 8 s and 9 s. Sperm motility was significantly higher when thawed at 60 °C for 8 s and 9 s (*p* > 0.05). There was no significant difference between the semen thawed at 8 s and 9 s, but the TMOT after 9 s was slightly lower than that after 8 s (47.86 vs. 52.65, respectively). Therefore, all subsequent semen dry ice preservation in this experiment was performed at 60 °C for 8 s.

### 3.2. Effect of Dry Ice Storage on the Quality of Frozen–Thawed Boar Semen

Frozen semen was transferred to dry ice for cryopreservation for 7 days and then re-transferred to LN. The effect of preservation of dry ice as a cold source on semen quality was examined on days 1, 3, 5, and 7, and then the semen was retransferred to LN. After the semen was frozen, its TMOT, MOT, and VCL decreased by more than 40% (92.71% vs. 53.43%, 85.62% vs. 46.39%, and 103.38% vs. 57.67%, respectively). Until day 7 of storage, there was no significant decrease in TMOT from days 0 to 3 (53.43% vs. 52.67% vs. 52.35%, respectively), but there was a reduction in TMOT after day 5 relative to day 0 (53.43% vs. 51.77% vs. 51.20%, respectively). Still, there was no significant difference (*p* > 0.05). No changes were observed in MOT or VCL, irrespective of the storage method (Table 2). Thereafter, on day 8, the three characteristics of the semen stored in LN decreased from those on day 0 (51.19% vs. 53.43%, respectively) and increased from day 7 (51.20% vs. 51.19%, respectively), but the results were not significant (*p* > 0.05).

Regarding the integrity of the spermatozoa acrosomes (Figure 2), significant differences were found between fresh and frozen semen in terms of acrosomes, plasma membrane, mitochondrial activity, and DNA integrity. There were no significant differences between preserving in LN, preserving in dry ice for 7 days, and retransferring to LN in terms of the effect on spermatozoa acrosomes (51.98% vs. 53.46% vs. 52.17%, respectively) (*p* > 0.05). With respect to the integrity of the plasma membranes of spermatozoa (Figure 3), there was almost no significant difference between dry ice and LN preservation on the spermatozoa plasma membrane (40.92% vs. 42.73% vs. 41.63%, respectively) (*p* > 0.05). In terms of the mitochondrial activity of the spermatozoa (Figure 4), no significant differences were observed regardless of whether spermatozoa were preserved in dry ice or LN (41.61% vs. 42.48% vs. 41.36%, respectively) (*p* > 0.05). The integrity of spermatozoa DNA (Figure 5) was not significantly impaired when semen was stored in LN or dry ice for 7 days or when it was retransferred to LN (47.64% vs. 47.11% vs. 47.69%, respectively) (*p* > 0.05).

### 3.3. Effect of Dry Ice Preservation on Antioxidant Function in Cryopreserved Boar Semen

From Table 3, it can be seen that the SOD and T-AOC of spermatozoa were significantly decreased (141.72 vs. 119.85; 2.31 vs. 1.33, respectively), MDA and ROS were increased considerably (4.54 vs. 7.31; 277.34 vs. 352.61, respectively), and antioxidant properties were significantly decreased after freezing compared to the fresh group (*p* < 0.05). Transferring frozen semen from LN to dry ice resulted in slightly lower SOD and T-AOC values for spermatozoa and significantly higher ROS and MDA values compared to LN preservation. When semen was retransferred from dry ice to LN on day 8, SOD and T-AOC values increased, and ROS and MDA values decreased, but the results were not significant (*p* < 0.05).

## 4. Discussion

Compared to fresh semen, the motility and vitality of frozen semen decreased significantly after thawing. This might be due to a series of physical and chemical shocks to the semen during the freezing process. For example, during the freezing–thawing process, semen underwent cellular dehydration, resulting in the production of ice crystals. This led to changes in spermatozoa function and damage to the membrane structure [22]. Boar spermatozoa, rich in unsaturated fatty acids, are susceptible to the effects of cold shock, and their viability and motility decrease by about 25–75% after cryopreservation [23]. In this experiment, cryopreserved semen showed around a 40% decrease in viability and motility after thawing.

The available literature on evaluating the impact of using dry ice as a cold source for semen transfer on porcine semen quality is scarce, and to the best of our knowledge, this is the first case of using dry ice for frozen semen transport in pigs. However, there are some studies in other species. Widely described as the most affected parameter was motility. In 1950, Polge [24] used dry ice as a cold source to freeze bull spermatozoa, which was kept in dry ice for 2–8 days and then thawed for artificial insemination. In total, 32 out of 38 inseminated cows were successfully impregnated, and 30 of them gave birth to calves. LN has gradually emerged as a better cold source than dry ice for the long-term preservation of semen [25]. However, using LN tanks for transportation has a high cost. The supply of LN is relatively limited in some areas, making it crucial to use dry ice instead of LN for freezing, storage, and transportation. Dry ice can also be used as an alternative cold source to LN for bovine artificial insemination under resource-scarce conditions. Unlike LN, dry ice does not require transfer cases made of unique materials for transportation, which reduces transportation costs. It is also less dangerous. Buranaamnuay [26] froze bovine semen and transferred it from LN to a −80 °C refrigerator, storing it for one month. He found that storing frozen semen at −80 °C did not have any effect on motility characteristics or morphology. In this study, we also found that dry ice was safe as an LN substitute during frozen semen transportation. Preservation of spermatozoa in dry ice did not significantly decrease motility. Seven days of storage in dry ice had no adverse effects on the motility of spermatozoa after thawing.

Acrosomal status and membrane integrity are another fundamental parameter of spermatozoa quality assessment tracked in our study. Studies have shown that transferring frozen horse semen from LN to dry ice for 7 days and then transferring it to LN had no adverse effect on its post-thaw characteristics [27]. However, some experiments have shown [4] that placing human semen in a −80 °C refrigerator can lead to a decrease in motility and vitality after thawing. This may be related to the differences in size and shape between spermatozoa. Additionally, it is related to the differences in the type and concentration of cryoprotectants. Long-term preservation of cells in dry ice increases the risk of ice crystal formation within the cell, resulting in damage to the spermatozoon membrane and thus disrupting the integrity of the cytoplasmic membrane. Still, this risk is reduced when stored in LN [28]. However, in this study, it was found that preserving frozen semen from pigs in dry ice for seven days and then retransferring it to LN had no significant effect on the spermatozoa plasma membrane.

Brotherton [29] reported that enzymes associated with cellular aging were largely ineffective below −70 °C. In dry ice preservation of human sperm, David [10] found no significant difference in mitochondrial activity when semen wheat tubes preserved in dry ice were returned to LN. In our study, we found that the mitochondrial activity of spermatozoa was significantly lower after freezing treatment compared to fresh semen and that freezing had a negative effect on spermatozoa mitochondria. This may be related to alterations in the spermatozoa plasma membranes due to cellular dehydration, high solute concentrations, and the recrystallization of spermatozoa during freezing, which resulted in altered sperm cell function and reduced mitochondrial activity [30,31]. Transferring sperm from LN to dry ice revealed a decrease in mitochondrial activity and DNA integrity compared to LN preservation, but the effect was not significant. This decrease was halted when the semen was retransferred to LN on the seventh day. This may be related to the nature of the cryoprotectant, which reaches a solid state at −75 °C and may be in a crystalline state. Still, in LN, there is less risk of crystal formation and subsequent cellular damage [32].

In summary, we show that dry ice is a reliable cold source and can be used for transporting or temporarily storing cryopreserved semen. However, there are few reports on the preservation of frozen porcine semen in dry ice for more than seven days. In our study, we demonstrated that frozen boar semen could be preserved in dry ice for around seven days and that sperm damage was not exacerbated when the semen was retransferred to LN. This provides new insight for the use of dry ice to transfer frozen semen in future domestic animal breeding strategies.

## 5. Conclusions

In the present study, we found that transferring frozen semen to dry ice did not cause damage to spermatozoa motility, plasma membrane integrity, acrosome integrity, mitochondrial activity, or DNA integrity. The dry ice-preserved semen was then retransferred to LN, and long-term preservation of porcine spermatozoa after transfer was also achieved. However, if we use dry ice to transfer a small amount of sample that cannot be put back into LN to be used, we should consider reducing the length of thawing to obtain higher-quality semen. When dry ice-preserved semen can be retransferred to LN again, we found that the quality of the semen is not affected. This means that dry ice has a positive role as a cold source in practical breeding in the future. It may also provide a suitable solution for the transfer of frozen semen in the future. Even so, we should pay special attention to other, more difficult-to-control factors: sufficient amounts of dry ice, suitable thawing temperatures, and the time of retransfer to LN are all factors that can cause sperm damage.

## Figures and Tables

**Figure 1 animals-14-01422-f001:**
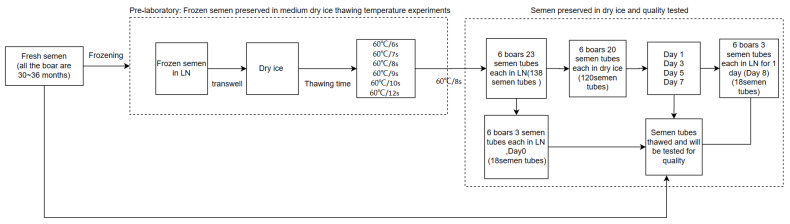
Flow diagram of the cryopreservation procedure steps and the experimental design. Arrows indicate when the semen analyses were performed during processing.

**Figure 2 animals-14-01422-f002:**
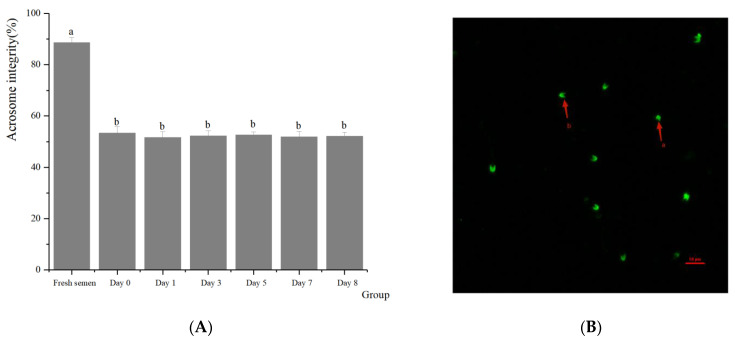
Effect of dry ice preservation on acrosome integrity in cryopreserved boar semen. (**B**) Spermatozoa stained with fluorescein isothiocyanate-labeled peanut agglutinin (FITC-PNA) with intact acrosomes (a. red arrowhead) or non-intact acrosomes (b. red arrowhead). (**A**) Percentage of spermatozoa with intact acrosomes. Values with different letters (mean ± S.E.M.) differ significantly (*p* < 0.05).

**Figure 3 animals-14-01422-f003:**
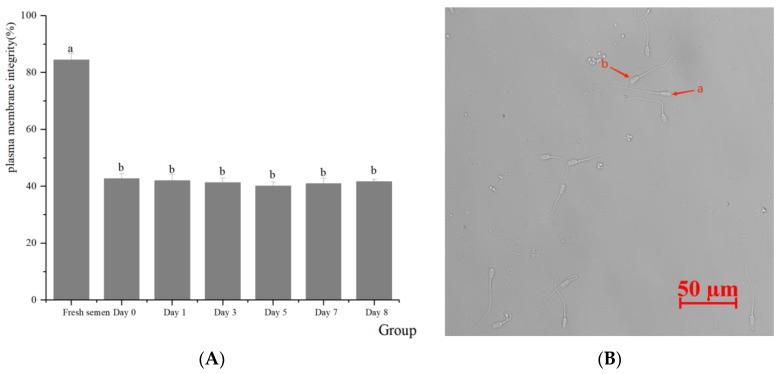
Effect of dry ice preservation on plasma membrane integrity in cryopreserved boar semen. (**B**)-a. arrows indicate positive HOST spermatozoa (intact plasma membranes, red arrow). (**B**)-b. Hypo-osmotic swelling test (HOST) and negative HOST spermatozoa (damaged plasma membranes, red arrow). (**A**) Percentage of viable spermatozoa with intact cell membranes. Values (mean ± S.E.M.) with different letters differ significantly (*p* < 0.05).

**Figure 4 animals-14-01422-f004:**
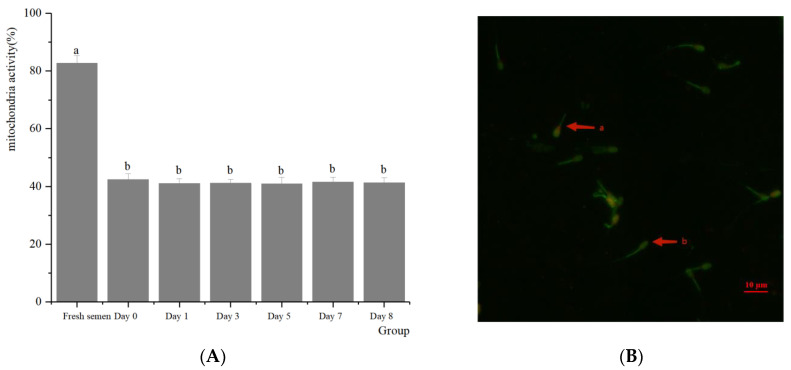
Effect of dry ice preservation on mitochondria activity in cryopreserved boar semen. (**B**) Spermatozoa mitochondrial activity assessed through staining with propidium iodide (PI) and rhodamine 123 (Rh 123). (**B**)-b. Arrows indicate intact mitochondrial membrane potential. (**B**)-a. Arrow indicates compromised mitochondrial membrane potential. (**A**) Percentage of semen mitochondrial activity. Values (mean ± S.E.M.) with different letters differ significantly (*p* < 0.05).

**Figure 5 animals-14-01422-f005:**
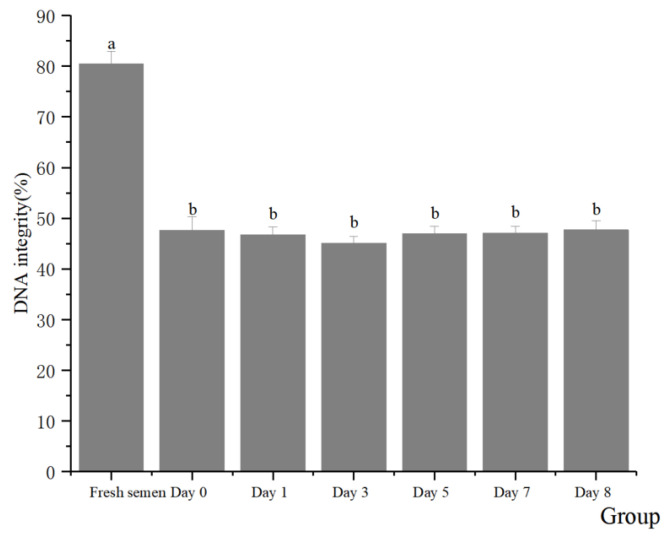
Effect of dry ice preservation on DNA integrity in cryopreserved boar semen. Note: The same letter or data with shoulder marks containing the same letter indicates a non-significant difference (*p* > 0.05), and a different lowercase letter indicates a significant difference (*p* < 0.05).

**Table 1 animals-14-01422-t001:** Effect of different thawing times on the motility characteristics of frozen semen preserved in dry ice.

Groups	TMOT/%	MOT/%	VSL/ (μm/s)	VCL/ (μm/s)	LIN/ %	STR/ %	VAP/ (μm/s)	WOB/%	ALH/ μm	BCF/Hz
60 °C/6 s	13.37 ± 2.23 ^d^	7.69 ± 3.17 ^d^	4.42 ± 1.37 ^d^	21.87 ± 1.98 ^d^	23.51 ± 1.37 ^c^	39.70 ± 1.53 ^c^	11.61 ± 2.92 ^d^	54.81 ± 1.95 ^c^	0.53 ± 0.05 ^e^	1.35 ± 0.41 ^d^
60 °C/7 s	37.85 ± 1.68 ^b^	30.51 ± 2.43 ^b^	15.92 ± 3.33 ^c^	46.75 ± 3.31 ^b^	35.17 ± 1.63 ^b^	63.71 ± 3.16 ^b^	24.41 ± 1.35 ^c^	53.39 ± 2.21 ^c^	0.96 ± 0.03 ^c^	3.37 ± 0.62 ^c^
60 °C/8 s	52.65 ± 3.19 ^a^	44.93 ± 3.31 ^a^	32.37 ± 1.75 ^a^	57.21 ± 3.52 ^a^	55.63 ± 2.85 ^a^	77.86 ± 1.67 ^a^	41.16 ± 2.17 ^a^	70.14 ± 2.28 ^a^	1.47 ± 0.04 ^a^	6.79 ± 0.51 ^a^
60 °C/9 s	47.86 ± 4.52 ^a^	39.86 ± 2.87 ^a^	27.65 ± 2.17 ^b^	55.17 ± 2.98 ^a^	51.13 ± 2.12 ^a^	73.28 ± 2.66 ^a^	35.61 ± 1.15 ^b^	63.94 ± 1.16 ^b^	1.24 ± 0.03 ^b^	5.17 ± 0.44 ^b^
60 °C/10 s	28.69 ± 2.41 ^c^	18.85 ± 3.37 ^c^	14.14 ± 2.57 ^c^	39.86 ± 2.97 ^c^	36.41 ± 1.56 ^b^	64.71 ± 1.78 ^b^	21.04 ± 1.07 ^c^	50.42 ± 2.51 ^c^	0.66 ± 0.02 ^d^	2.53 ± 0.37 ^c^
60 °C/12 s	7.76 ± 3.17 ^e^	4.13 ± 2.44 ^d^	2.29 ± 0.78 ^d^	11.36 ± 1.43 ^e^	21.75 ± 1.14 ^c^	36.56 ± 1.23 ^c^	6.51 ± 0.73 ^e^	55.73 ± 2.64 ^c^	0.47 ± 0.03 ^e^	1.44 ± 0.25 ^d^
*p* value	<0.01	<0.01	<0.01	=0.059	=0.003	<0.01	<0.01	=0.003	<0.01	=0.004

Note: The following motility patterns were analyzed [20,21]: total motility (MOT; %), progressive motility (PMOT; %), curvilinear velocity (VCL; mm/s), average path velocity (VAP; mm/s), straight-line velocity (VSL; mm/s), straightness (STR: VSL/VAP; %), linearity (LIN: VSL/VCL; %), wobble (WOB: VAP/VCL; %), lateral head displacement (ALH; mm) and beat cross frequency (BCF; Hz). LIN = VSL/VCL × 100; STR = VSL/VAP × 100; WOB = VAP/VCL × 100. No letters in the same column of data or data with shoulder markers containing the same letter indicate that the difference is not significant (*p* > 0.05), and different lowercase letters indicate that the difference is significant (*p* < 0.05). The following table is the same.

**Table 2 animals-14-01422-t002:** Effect of dry ice storage on motility of frozen–thawed boar semen.

Groups	TMOT/%	MOT/%	VSL/ (μm/s)	VCL/ (μm/s)	LIN/ %	STR/ %	VAP/ (μm/s)	WOB/%	ALH/ μm	BCF/ Hz
Fresh semen	92.71 ± 2.31 ^a^	85.62 ± 1.75 ^a^	75.57 ± 2.03 ^a^	103.38 ± 1.77 ^a^	85.61 ± 2.44 ^a^	94.85 ± 3.90 ^a^	79.76 ± 3.14 ^a^	90.45 ± 2.49 ^a^	4.67 ± 0.17 ^a^	13.18 ± 0.54 ^a^
Day 0	53.43 ± 1.14 ^b^	46.39 ± 1.04 ^b^	32.97 ± 2.52 ^b^	57.67 ± 2.12 ^b^	52.97 ± 2.05 ^b^	72.51 ± 3.11 ^b^	42.56 ± 1.25 ^b^	72.13 ± 1.67 ^b^	2.61 ± 0.41 ^b^	7.44 ± 0.39 ^b^
Day 1	52.67 ± 1.74 ^b^	45.71 ± 2.62 ^b^	30.59 ± 1.55 ^b^	55.96 ± 2.67 ^b^	51.76 ± 2.34 ^b^	71.44 ± 2.17 ^b^	40.93 ± 2.92 ^b^	71.90 ± 1.79 ^b^	2.27 ± 0.28 ^b^	7.35 ± 0.56 ^b^
Day 3	52.35 ± 1.97 ^b^	44.17 ± 1.46 ^b^	31.46 ± 2.16 ^b^	56.81 ± 2.16 ^b^	50.71 ± 1.88 ^b^	71.71 ± 1.93 ^b^	41.43 ± 1.89 ^b^	71.77 ± 1.02 ^b^	2.23 ± 0.20 ^b^	7.37 ± 0.43 ^b^
Day 5	51.77 ± 1.36 ^b^	45.87 ± 2.46 ^b^	31.12 ± 2.13 ^b^	56.17 ± 1.99 ^b^	51.37 ± 2.31 ^b^	70.93 ± 2.75 ^b^	40.66 ± 1.49 ^b^	69.83 ± 2.79 ^b^	2.41 ± 0.53 ^b^	7.26 ± 0.37 ^b^
Day 7	51.19 ± 1.46 ^b^	45.69 ± 1.43 ^b^	31.63 ± 1.43 ^b^	56.65 ± 1.69 ^b^	51.69 ± 1.73 ^b^	71.39 ± 1.59 ^b^	40.87 ± 1.35 ^b^	70.56 ± 2.13 ^b^	2.37 ± 0.42 ^b^	7.33 ± 0.39 ^b^
Day 8	51.20 ± 1.33 ^b^	45.13 ± 1.44 ^b^	30.76 ± 2.31 ^b^	56.93 ± 2.59 ^b^	51.98 ± 1.75 ^b^	71.86 ± 2.14 ^b^	41.04 ± 1.97 ^b^	71.31 ± 1.45 ^b^	2.43 ± 0.29 ^b^	7.27 ± 0.39 ^b^

Note: No letters in the same column of data or data with shoulder markers containing the same letter indicate that the difference is not significant (*p* > 0.05), and different lowercase letters indicate that the difference is significant (*p* < 0.05).

**Table 3 animals-14-01422-t003:** Effect of short-term storage in dry ice on oxidative indices of spermatozoa.

Groups	SOD (U/mL)	MDA (nmol/L)	ROS (RFUs)	AOC (nmol/L)
Fresh semen	141.72 ± 3.56 ^a^	4.54 ± 0.38 ^b^	277.34 ± 1.29 ^c^	2.31 ± 0.36 ^a^
Day 0	119.85 ± 2.78 ^b^	7.31 ± 0.57 ^a^	352.61 ± 1.96 ^b^	1.53 ± 0.51 ^b^
Day 1	117.93 ± 3.51 ^b^	7.55 ± 0.72 ^a^	364.51 ± 2.32 ^a^	1.47 ± 0.33 ^b^
Day 3	118.13 ± 3.16 ^b^	7.41 ± 0.67 ^a^	358.03 ± 3.47 ^a^	1.41 ± 0.29 ^b^
Day 5	118.69 ± 4.14 ^b^	7.57 ± 0.71 ^a^	360.44 ± 2.07 ^a^	1.35 ± 0.31 ^b^
Day 7	117.79 ± 3.67 ^b^	7.63 ± 0.59 ^a^	366.74 ± 3.99 ^a^	1.39 ± 0.27 ^b^
Day 8	118.13 ± 2.98 ^b^	7.39 ± 0.44 ^a^	361.97 ± 2.67 ^a^	1.49 ± 0.28 ^b^

Note: No letters in the same column of data or data with shoulder markers containing the same letter indicate that the difference is not significant (*p* > 0.05), and different lowercase letters indicate that the difference is significant (*p* < 0.05).

## Data Availability

The original contributions presented in the study are included in the article, further inquiries can be directed to the corresponding authors.

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
