# Peer review of "Evaluation of Dry Ice for Short-Term Storage and Transportation of Frozen Boar Semen"

_animals, 2024, doi:10.3390/ani14101422_

Round 1
Reviewer 1 Report
Comments and Suggestions for Authors
The present study evaluated the feasibility of storing frozen boar semen in dry ice for short term with an objective to overcome the security concern about transportation of cryopreserved boar semen in liquid nitrogen. The results of the present study demonstrated that no apparent damage was induced by storage in dry ice in terms of sperm motion parameters, morphology and antioxidant indexes, compared to that of constant storage in liquid nitrogen. This study provides a useful method to enhance the local and international distribution of frozen boar semen, which will benefit the porcine industry for facilitating communication in genetic improvement at both national and international levels. There are several points regarding the contents of this study I would like to share my opinions with the authors.
General comments:
This study showed a good presentation of the results. In relation to the research design, what are the races of the boars that provided frozen semen samples for this study? Was there any influence of boar race observed? Were the boars and frozen straws selected? How were they selected? It is recommended to provide information on the freezing protocol that was used to make frozen semen for this study. Furthermore, no thawing protocol for liquid nitrogen storage was provided. Did the authors use the same thawing protocol for dry ice and liquid nitrogen storage? Besides, concerning the thawing protocol for storage in dry ice, combination with high temperature (60 ℃) and short time (8s) showed the best results. However, it is pretty difficult to control the time. As the results showed, 1s led to significant changes. Why did not the authors consider lower temperature like 37 ℃ but longer time, like 30 s for thawing?
Specific comments:
Q1. Line 27 and 28, errors in P value should be corrected, which should be 0.05 instead of 0.5. Line 93, please recheck the information of the company that produce AO reagent.
Q2. Line 166-186, it is suggested to point out the results expression of MDA and ROS, and the equation for calculating SOD activity.
Q3. Line 191, what is the type of post hoc analysis?
Q4. In table 1, line 205, for those parameters that showed significant differences between groups, there are no letters indicating the difference, e.g. VSL, LIN, STR, VAP, WOB, ALH and BCF.
Q5. Line 212, in the title, according to the experiment design, it was not frozen-thawed semen that were transferred from LN to dry ice, but frozen semen. Please recheck this information and make corrections.
Q6. For table 1 and 2, it is recommended to use histogram or line chart to present the results, which will be more straight to get the information.
Q7. How long were the frozen semen stored in liquid nitrogen after transferring back from dry ice at day 8?
Comments on the Quality of English LanguageMinor
Reviewer 2 Report
Comments and Suggestions for Authors
The paper addresses the possibility of using dry ice as an alternative for the transportation of frozen boar semen samples. In this light, the rationale is interesting and could be exploited further in the breeding practice. Nevertheless, the manuscript presents with issues that need to be carefully addressed and revised in order to increase the scientific soundness and presentation quality.
What hits first, is that the presentation quality is very sloppy. The authors need to re-visit the Instructions for Authors and revise the paper accordingly. The references are not cited according to the Instructions, they are referenced in superscripts in some parts of the manuscript. Some of the tables and figures are missing the legends. At the same time, some of the methods are described in past tense, whereas others are put as a literal citation of the protocol – this needs to be unified.
The Material and Methods section is very confusing. It seems that the study focuses more on the time-dependent effects of the thawing conditions rather than the actual comparison of liquid nitrogen versus dry ice. The experimental design is very confusing, the authors mention that some samples were transferred back to liquid nitrogen, some were not, which, combined with different days of storage and thawing times is very difficult to comprehend. I strongly suggest adding a figure depicting the actual experimental design for more clarity.
It is not clear what was done with the semen samples. How may boars were used? What breed of boars was included? What was the age of the boars? How was semen collected. More importantly – how were the ejaculates initially frozen?
A lot of the methods are missing the manufacturers (city and state) of the chemicals and instrumentation. Methods related to the antioxidant status are very vaguely described, making the methodology irreproducible. What types of samples were used – whole semen? Spermatozoa?
More importantly, I feel like the design misses another control. Fresh semen serves as the negative control (however the paper does not describe how the sample was processed), however samples stored in liquid nitrogen exclusively should serve as a positive control. Comparing samples stored using dry ice with fresh semen exclusively does not answer the question asked in the study – is dry ice as effective as liquid nitrogen? If the question is whether dry ice can assure a time-dependent preservation of the sperm quality, then the statistics needs to be re-done by using two-way ANOVA, taking the different groups, storage and/or thawing time into consideration.
The results section does not need to have the obtained values for the parameters to be included in the text if there are Tables depicting the same values.
Finally, the Discussion needs more attention. The study does not really interpret the data, it only serves as a summary of reports that have been previously published on this topic. Finally, limitations, prospects, and the contribution of the study to future improvement on this issue in practical breeding needs to be discussed.
Comments on the Quality of English Language
I recommend the authors to carefully revise the linguistic aspect of the paper as there are some grammar and/or typo errors throughout the text.
Author Response
请参阅附件

Reviewer 3 Report
Comments and Suggestions for Authors
General comments:
This study aimed to evaluate the effect of dry ice on transport of boar (straw) semen up to 8 days. The subject is actual and this work fit well the scope of this journal. According to Tables and figures, the results are apparently reliable (presenting scientific soundness). However, the scientific writing is too confusing. The writing of this manuscript should be sequential and consistent to improve the readability of the manuscript. This is particularly evident in the introduction and M&M sections. It is not possible to determine what is the effective study design; moreover, in results section, the thawed semen was compared with fresh semen (?). Is the comparison between groups made between thawed semen directly from liquid nitrogen (control group) and from dry ice at different times? Why it was reported that the semen is transferred again to liquid nitrogen? And what straws? Is the fresh semen evaluated, cryopreserved in liquid nitrogen, placed in dry ice for different periods, and evaluated for a new evaluation? And the 6 s, 7 s, 8 s, 9 s, 10 s, and 12 s of thawing is for d0, 1 d, 3d, 5 d, 7 d and 8 d (6x5 x 3 replicates=90?). I suggest to clarify the study design for a new re-assessment of this manuscript to make an adequate peer-review. Probably an English language edition can improve the grammar and sequence of ideas, and as consequent the readability, even partially.
Specific comments:
L15-18: Please re-write the simple summary.
L39: In china.
L44-46: I suggest to insert a reference.
L53: Structure of spermatozoa.
L63-65: Really? I think that this statement is not correct. Containers with LN are widely used to transport semen. What are the limitations for boar semen (volume?) Please justify adequately your study.
L77: In cows.
L88: Why to insert this here? Please move it to 2.3.6.
L94: It is very confusing this part. I cannot understand when the semen was effectively evaluated in each group.
L95: 0.5 mL of volume?
L95-106: The description is confusing. Please re-write.
L114: This is a repetition.
L202: In these situations, use “respectively”: (47.86 vs. 52.65, respectively). Please check the manuscript for this issue.
L205: please insert the superscript letters by alphabetic order in each column.
L225: Each table or figure is independent. Please insert the respective caption.
Round 2
Reviewer 2 Report
Comments and Suggestions for Authors
The authors have properly addressed all my questions and have very much improved their manuscript in the newly revised version.
In my opinion the manuscript is now suitable for publication.
Author Response
It's a great honor to have your endorsement of this work, and thank you for your questions, the article is a great help to the quality of the article.
Reviewer 3 Report
Comments and Suggestions for Authors
General comments:
Dear Authors, thanks for providing this revised version. The comments on original versions were taken in account. The study design is now clarified. The results were adequately discussed. Some minor issues were found and should be corrected. Please use a space before the references in the whole text. In Table 1, I suggest to reorder (top to bottom) the superscript letters for each column (a,b,c,..”). The conclusion needs to be re-written without unnecessary repetitions and information.
Specific comments:
L140: Please reorder the figures. This one is the Fig. 1. (thanks for the insertion of this figure).
L274: “… spermatozoa acrosomes…”. Spermatozoa are cells; semen = spermatozoa + fluid. In this manuscript the term “semen” is inappropriately used (in several cases) as synonymous of spermatozoa or sperm cells. Please pay attention for this issue in the whole manuscript (including in the legend of figures)
L311: “…liquid nitrogen…”. Please use the abbreviation “LN”. Check the text.
L313: “… a high cost.”.
L356-373: This part can serve as conclusion after re-write it (please use the past tense regarding your findings). Avoid repetitions.
L357-358: I suggest to remove this sentence.
L364-373: You cannot start your conclusion with “however”. Please re-write the conclusion according the main findings of this study.
